# Protective Effects of Hydrogen-Rich Saline Against Hemorrhagic Shock in Rats via an Endothelial Glycocalyx Pathway

**DOI:** 10.3390/biomedicines13040833

**Published:** 2025-03-31

**Authors:** Aya Kimura, Koichi Suehiro, Tokuhiro Yamada, Yasuda Shinta, Takashi Juri, Yohei Fujimoto, Shinichi Hirano, Takashi Mori

**Affiliations:** 1Department of Anesthesiology, Graduate School of Medicine, Osaka Metropolitan University, Osaka 545-8586, Japan; h21058y@omu.ac.jp (A.K.); t-yamada@med.osakacity-hp.or.jp (T.Y.); 5v00py.32@gmail.com (Y.S.); takataka0061@yahoo.ne.jp (T.J.); fujimotoyohei@gmail.com (Y.F.); takashimoritakashi@gmail.com (T.M.); 2Independent Researcher, 5-8-1-207 Honson, Shibasaki, Numazu 253-0042, Japan; hirano_0719@yahoo.co.jp

**Keywords:** hydrogen, hemorrhagic shock, fluid resuscitation, endothelial glycocalyx

## Abstract

**Background/Objective:** The endothelial glycocalyx is a gel-like layer on the vascular endothelial surface that is crucial for maintaining vascular homeostasis. Massive bleeding leads to the shedding of the glycocalyx, which can lead to vascular leakage during fluid administration. Recently, the beneficial effect of hydrogen inhalation in the treatment of hemorrhagic shock has been reported. However, the efficacy of hydrogen-rich saline in protecting the glycocalyx remains unclear. In this study, we investigated the effects of hydrogen-rich saline on glycocalyx degeneration. **Methods:** Rats under general anesthesia were divided into five groups: the sham, hemorrhagic shock, normal saline, colloid solution, and hydrogen-rich saline groups (*n* = 6 for each group). Blood was withdrawn, and blood pressure was maintained at 30–35 mmHg for 60 min. After inducing hemorrhagic shock in this way, each infusion product was administered intravenously to maintain blood pressure at 80 mmHg for 60 min. Glycocalyx thickness was assessed using the GlycoCheck system. **Results:** The use of hydrogen-rich saline significantly improved the survival rate (*p* < 0.05), and glycocalyx degeneration was significantly suppressed (*p* < 0.001), indicating the protective effect of hydrogen on the glycocalyx. **Conclusion:** Intravenous administration of hydrogen-rich saline in hemorrhagic shock attenuates glycocalyx degeneration compared to conventional fluid resuscitation, which can improve survival rates.

## 1. Introduction

Massive hemorrhage is one of the most devastating causes of preventable death in the perioperative period [1]. Fluid administration is the first strategy when treating massive hemorrhage. However, excessive fluid administration, particularly during the perioperative period, can increase not only circulatory and respiratory complications but also gastrointestinal complications, such as ileus and suture failure. Hemorrhage-induced organ damage is attributed to hypoxia, hypoperfusion, and cell death due to decreased ATP supply. In addition, while reperfusion after correction of the circulatory failure improves oxygenation, it simultaneously generates large amounts of reactive oxygen species (ROS), resulting in calcium overload and the induction of cell apoptosis. Consequently, when treating massive hemorrhage, reperfusion injury must be reduced after the improvement in circulatory failure.

The first choice of fluid product for treating massive hemorrhage is crystalloids. However, the administration of large volumes of crystalloid is associated with increased risk of acute respiratory distress syndrome, multiple organ dysfunction syndrome, compartment syndrome, dilutional coagulopathy, and mortality in patients with traumatic hemorrhagic shock [2]. Previous studies have indicated that crystalloid-based resuscitation is inferior to that of albumin and fresh-frozen plasma when treating massive hemorrhage [3,4]. Although blood transfusion may be beneficial for patients with critical hemorrhagic shock, some risks are associated with the use of blood products. The availability of blood products is limited and depends on the amount of storage in the hospital. Additionally, blood transfusion can cause some reactions, such as allergic reactions, transfusion-related acute lung injury (ALI), and transfusion-associated circulatory overload [5,6]. In these respects, more effective infusion products are needed for treating massive hemorrhage.

The maintenance of vascular permeability is a new concept and is an issue of great importance for effective fluid resuscitation. The endothelial glycocalyx plays an important role in maintaining vascular permeability [7]. The endothelial vascular surface is covered with glycocalyx, a ubiquitous gel-like layer consisting of a membrane-binding domain including proteoglycans, glycosaminoglycan side-chains, and plasma proteins [8]. The glycocalyx has various roles, including the maintenance of microvascular permeability, regulating microvascular tone, preventing thrombosis, and controlling leukocyte adhesion. The endothelial glycocalyx layer is markedly fragile [8]. It is easily damaged by ischemia–reperfusion injury, inflammation, trauma, hyper- or hypovolemia, atherosclerosis, and diabetes [9]. Massive hemorrhage also causes the degeneration of the glycocalyx layer [10]. When the endothelial glycocalyx is destroyed, capillary leakage, tissue edema, accelerated inflammation, platelet aggregation, and coagulation disorder ensue, leading to various complications [8,9] Therefore, the maintenance of the endothelial glycocalyx is a key issue when treating massive hemorrhage.

Hydrogen (H_2_) is a stable molecule that has various biological advantages, including antioxidative, anti-inflammatory, anti-apoptotic, and antitumor activities. Because of its small molecular size, H_2_ spreads rapidly and penetrates the blood–brain barrier, cell membranes, and mitochondrial membranes, reaching the site of the lesion and exerting its effects [11]. There are some ways to provide H_2_ into body, such as inhalation, intravenous infusion, oral take, and cutaneous exposure. Previous studies showed that H_2_ gas inhalation decreased oxidative stress and prevented the degeneration of the glycocalyx layer even after hemorrhagic shock in a rat model [12,13] However, the efficacy of H2-rich saline on glycocalyx protection in a hemorrhagic shock model is not fully investigated.

If H_2_-rich saline could decrease the degeneration of the endothelial glycocalyx during hemorrhagic shock, it might be more effective as an infusion compared to the currently available infusions for treating patients with massive hemorrhage.

In this experimental study, we assessed the effect of intravenous administration of H_2_-rich saline on glycocalyx protection in a rat hemorrhagic shock model.

## 2. Materials and Methods

All experimental protocols of this study were approved by the Institutional Animal Care and Use Committee of our institution (Number: 20019, Osaka Metropolitan University Graduate School of Medicine, Osaka, Japan). Experimental procedures were conducted in accordance with the ARRIVE guidelines developed by the National Center for the replacement, refinement, and reduction of animals in research.

This study consisted of three experiments: In Experiment 1, we assessed the protective effect of H_2_-rich saline against hemorrhagic shock from the point view of the endothelial glycocalyx, cardiac contractility, and blood data. Experiment 2 evaluated the histopathological outcomes after hemorrhagic shock. Experiment 3 compared the survival rate between hemorrhagic shock model animals treated with conventional fluid resuscitations or with H_2_-rich saline resuscitation.

### 2.1. Animals and Surgical Preparation

Sixty male 10–12-week-old Sprague–Dawley rats weighing 450–550 g (Clea Inc., Osaka, Japan) were housed in plastic cages at the animal center of our institution with 12 h/12 h dark/light cycles. Food and water were freely available. All experimental procedures were conducted during the daytime. Rats were anesthetized by using 5% sevoflurane (NIKKO Pharmaceutical Co., Ltd., Gifu, Japan) inhalation and 40 mg/kg pentobarbital (Sigma-Aldrich, St. Louis, MO, USA) intraperitoneal administration. Anesthesia was maintained with oxygen, nitrogen, and 3% sevoflurane (total flow rate 4 L/min) via an anesthesia machine (Aestiva 5-7100; Datex-Ohmeda Inc., Madison, WI, USA). After anesthesia induction, an arterial catheter was placed in the left carotid artery for continuous left ventricular pressure monitoring (Millar Mikro-Tip UM-320P; Millar, Houston, TX, USA). A central venous line was placed in the right femoral vein (PE-50: Becton, Dickinson and Company, Franklin Lakes, NJ, USA) for fluid administration. Systemic heparinization was applied. Tracheotomy was performed with a tracheal cannula (16-gauge intravenous cannula [Insyte-W; Becton, Dickinson and Company, Singapore]). Rats were connected to a ventilator (SN480-7; Shinano Co., Tokyo, Japan), with the tidal volume set to 10 mL/kg to manage the partial pressure of carbon dioxide between 35 and 40 mmHg. The mesentery and right quadriceps femoris were exposed to observe the microvascular endothelial glycocalyx using the GlycoCheck (Microvascular Hearth Solutions Co, Oceanside, CA, USA) system. Body temperature was measured using a rectal thermometer (DCBcIII; Technowood, Co., Ltd., Tokyo, Japan).

### 2.2. Study Protocol

After induction and surgical preparation, rats were randomly divided into five groups using sealed envelopes: sham, shock (with only hemorrhagic shock induced, without resuscitation), NS (hemorrhagic shock + fluid resuscitation with normal saline), HES (hemorrhagic shock + fluid resuscitation with colloid), and H_2_-NS (hemorrhagic shock + fluid resuscitation with H_2_-rich saline). Similar to a previous study [10], to induce hemorrhagic shock in rats, blood was withdrawn until a mean arterial pressure (MAP) of 30–35 mmHg was achieved and maintained for 1 h. After hemorrhagic shock induction, fluid resuscitation was performed for rats in the NS, HES, and H_2_-NS groups using the respective abovementioned solutions. Fluid was infused via the right femoral vein over 15 min to maintain a MAP of 80 mmHg (Figure 1).

In Experiment 1, the glycocalyx was observed at baseline (60 min before fluid resuscitation) and at 30, 60, 90, and 120 min after resuscitation. The dP/dt, the amount of ROS, and arterial blood gas were measured at baseline and at 30, 60, and 120 min after resuscitation. In Experiment 2, 30 rats were euthanized 2 h after fluid resuscitation. In Experiment 3, survival rates were observed at 6 h after resuscitation.

### 2.3. H2-Rich Saline

We prepare the H_2_-rich saline by developing a simple disposable type of device (MiZ Co., Ltd., Kamakura, Japan). This device consists of a H_2_-generating agent and a polyethylene bag enclosed within an aluminum bag. We wet the H_2_-generating agent with a small amount of water and placed both the polyethylene saline bag and the H_2_-generating agent inside the aluminum bag at a gauge pressure of 0.4 Mpa, as previously described [14]. Thus, 100% H2 was generated within the aluminum bag. Following vacuum treatment, H_2_ could aseptically dissolve into the biocompatible solution through the polyethylene bag containing the saline solution. Using this protocol, we managed to maintain the H_2_ concentration at approximately 1.5 ppm.

### 2.4. Glycocalyx Analysis

The thickness of the endothelial glycocalyx in the mesentery and quadriceps femoris was measured using the GlycoCheck system (Figure 2). The system captures images of capillaries in real-time and measures their thickness using the built-in software. Glycocalyx thickness was assessed to measure the perfusion boundary region (PBR). As the impermeable layer of the glycocalyx decreases, the width of erythrocyte perfusion in the vascular lumen expands outward, leading to an increase in PBR values.

### 2.5. Cardiac Contraction

Cardiac contraction and left ventricular pressure (LVP) were continuously measured by using the arterial catheter (Millar Mikro-Tip, Pearland, TX, USA). From the pressure values, dP/dt was also calculated to evaluate cardiac contractility.

### 2.6. Blood Analysis

Arterial blood was collected from the left carotid artery at baseline (60 min before fluid resuscitation) and at 30 min and 120 min after fluid resuscitation. pH, base excess (BE), PaO2, PaCO2, sodium, potassium, hemoglobin, lactate, blood urine nitrogen (BUN), and creatinine were measured using a blood gas analyzer (iSTAT; Abbott Point Care Inc. Princeton, NJ, USA).

### 2.7. Reactive Oxygen Species

To assess the antioxidant effect of H2-rich saline, we conducted the derivatives-reactive oxygen metabolites (d-ROMs) test at baseline and at 30 min and 120 min after hemorrhagic shock induction. This test measures the amount of hydroperoxides and free radical metabolites in the blood. Previous studies have measured the amount of hydroperoxides with the d-ROMs test using a FREE Carpe Diem device (Wismerll Co., Ltd., Tokyo, Japan) [15,16,17]. Hydroperoxides are unstable intermediates formed by the oxidation of lipids, peptides, and amino acids, and their levels are comparable to those of free radicals [15,16]. For this analysis, blood samples were collected, and the serum was separated from the blood via centrifugation. Recent study indicated that the d-ROMs test and biological antioxidant potential have a correlation with oxidative stress [18]. Twenty serum samples were then added to an acetate-buffered solution (pH 4.8) to generate new radicals. These radicals caused a color change in the solution, from colorless to magenta. The intensity of the magenta color, measured spectrophotometrically at 505 nm, reflected the amount of free radicals present.

### 2.8. Histopathological Analysis

Two hours after fluid resuscitation, 30 rats were euthanized with 8.0% sevoflurane for 10 min (Experiment 2). The left lung was removed, and Ringer’s solution (100 mL) and 10% formalin (150 mL) were perfused from the pulmonary artery. Then, the left upper lobe was removed and fixed in 10% formalin for 72 h. Thereafter, a segment of the left upper lobe was dissected and embedded in paraffin. The left upper lobe was sectioned and stained with hematoxylin and eosin. The section was observed at 20× magnification under a light microscope (BX53; Olympus Co., Tokyo, Japan). A histopathological evaluation was performed by using the average acute lung injury (ALI) scores from three different slides, according to previously reported criteria [19]. The evaluation criteria consisted of eight items: edematous alveolar wall, infiltration or inflammation in the air space, alveolar wall, vessel wall, alveoli atelectasis, necrosis, hemorrhage, and hyaline membrane formation. Each item was graded on a scale of 0–4 (0, no injury; 4, injury throughout the field) at 20× magnification with a light microscope (BX53; Olympus Co.). The histopathological evaluation was performed on 12 different areas from each slide.

### 2.9. Survival Rate

At 6 h after hemorrhagic shock, we assessed the survival rate in each group (Experiment 3).

### 2.10. Statistical Analysis

We hypothesized the PBR change in the NS and H_2_-NS groups as 200% and 150%, respectively, with a standard deviation of 25%. We assumed that a power of 80% and an alpha risk of 0.01 were sufficient for multiple comparisons. Accordingly, a sample size of 6 animals per group was calculated.

The Brown–Forsythe test and the Shapiro–Wilk test were used to assess the equality of variance and normality, respectively. To evaluate PBR, the values before hemorrhage shock (at baseline) were defined as 100%. Variables were measured at 30 min intervals and were represented as the percentage change from baseline. These values were compared among groups using two-way repeated-measures analysis of variance (ANOVA) with Tukey’s post hoc test for multiple comparisons. The dP/dt and the amount of ROS were also analyzed using two-way repeated-measures ANOVA with the Tukey post hoc comparison test. The ALI scores were compared using one-way repeated-measures ANOVA with Tukey’s post hoc multiple comparison test. Blood test values were represented as the mean and standard deviation. The comparisons of these variables were performed using one-way repeated-measures ANOVA with Tukey’s post hoc multiple comparison test. The Kaplan–Meier curved and log-rank tests were used to identify the difference in the survival rates among groups.

Statistical analyses were performed using StatFlex version 6.0 (Artech. Co., Ltd., Osaka, Japan) and the SigmaPlot 13.0 software (Systat Software Inc., San Jose, CA, USA). All *p* values were adjusted with Tukey’s post hoc test for multiple comparisons, and an adjusted *p* < 0.05 was considered statistically significant.

## 3. Results

### 3.1. Characteristics of Rats

No significant differences in body weight or blood loss was seen among the groups (Table 1). However, in the H_2_-NS group, the amount of fluid required for resuscitation was significantly lower than that in the NS group (*p* < 0.001).

### 3.2. Effect of H_2_ on Glycocalyx Shedding

Figure 3A shows the changes in glycocalyx thickness in the mesentery in the sham, shock, NS, HES, and H_2_-NS groups. The time-course analysis with two-way repeated-measures ANOVA showed significant differences among the groups (F[3,80] = 613.2, *p* < 0.001). The glycocalyx thickness in the mesentery was significantly better maintained in the H_2_-NS group after hemorrhagic shock than in the shock and NS groups at each time point (*p* < 0.001). Figure 3B indicates the changes in the glycocalyx thickness in the quadriceps in the sham, shock, NS, HES and H_2_-NS groups. The result of two-way repeated-measures ANOVA for the glycocalyx in the quadriceps was also significantly different among the groups (F[3,80] = 411.4, *p* < 0.001). The glycocalyx thinning in the quadriceps was also significantly attenuated in the H_2_-NS group compared to that in the shock, NS, and HES groups at all time points (Figure 3B).

### 3.3. Effect of H_2_ on Cardiac Contraction

Figure 4 shows the changes in dP/dt throughout the protocol. Two-way repeated-measures ANOVA indicated that the time course of change in cardiac contraction was significantly different among the groups (F[3,80] = 171.3, *p* < 0.001). At 1 and 2 h after hemorrhagic shock induction, cardiac contraction was significantly maintained in the H_2_-NS group compared to the shock, NS, and HES groups (*p* < 0.001).

### 3.4. Effect of H_2_ on ROS Levels

Figure 5 shows changes in ROS levels after shock induction. The time-course change using two-ways repeated-measures ANOVA was significantly different among the groups (F[3,80] = 55.7, *p* < 0.001). At 2 h after the shock phase, the increase in ROS in the H_2_-NS group was significantly suppressed compared to that in the shock, NS, and HES groups (*p* < 0.001).

### 3.5. Effect of H_2_ on Lung Histopathology

Histopathological changes in lung tissue after hemorrhagic shock are shown in Figure 6A. The lung in the sham group demonstrated normal tissue histology, whereas the shock, NS, and HES groups showed damaged tissue: alveolar and interstitial edema, inflammation, and the exudation of red blood cells. Atelectasis and hyaline membrane formation were also observed in parts. The ALI score differed significantly between the H_2_-NS group and the other fluid resuscitation groups (*p* < 0.001). On the other hand, the ALI score did not differ significantly between the NS group and the shock group (Figure 6B).

### 3.6. Effect of H2 on Survival Rate

The survival rate at 6 h after resuscitation is shown in Figure 7. The survival rate in the shock, NS, and HES groups was 0%, 0%, and 17%, while it was 83% and 100% in the H_2_-NS and sham groups, respectively. Rats in the H2-NS group demonstrated a significantly higher survival rate than those in the shock and NS groups (*p* < 0.05).

### 3.7. Blood Analysis

The results of the blood gas analysis at baseline and at 30 min and 120 min after fluid resuscitation are shown in Table 2, respectively. No significant difference in blood gas analysis was seen among the groups at baseline. However, in the H2-NS group, lactic acidosis was significantly improved compared to that in the shock group at both 30 and 120 min after resuscitation (*p* < 0.001). The H_2_-NS group showed no significant adverse effects on BUN or creatinine levels. The PaO_2_ in the H2-NS group was significantly maintained after fluid resuscitation (*p* < 0.001). Hemoglobin levels at 2 h after fluid resuscitation in the H_2_-NS group were significantly different from those in the other fluid groups (*p* < 0.05).

## 4. Discussion

In the current study, we have demonstrated the protective effect of H_2_-rich saline on endothelial glycocalyx degeneration in a hemorrhagic shock model. H_2_-rich saline contributed to beneficial outcomes in terms of cardiac contraction, ROS levels, ALI scores, and survival rate.

Massive hemorrhage in the perioperative period carries a high mortality rate and represents a significant clinical challenge. The causes of hemorrhagic shock vary widely and include trauma, maternal hemorrhage, gastrointestinal bleeding, perioperative hemorrhage, and the rupture of an aneurysm [1]. Fluid infusion is essential for successful resuscitation in massive hemorrhage cases. Isotonic fluids are typically administered from the initiation of resuscitation. However, improving the shock state often necessitates the administration of large volumes of fluid, which may cause some complications. While blood transfusion products are effective for treating massive hemorrhage, their availability is limited. Therefore, more effective and highly available fluid resuscitation products are needed for the management of hemorrhage. Previous studies have indicated that crystalloid-based resuscitation is inferior to that of albumin and fresh-frozen plasma when treating massive hemorrhage [3,4].

The glycocalyx on the vascular endothelial surface regulates vascular barrier function. When it is damaged, proteins are extravasated; blood flow is lost, and platelet and leucocyte adhesion increases. The endothelial glycocalyx is damaged by ischemia–reperfusion injury after hemorrhagic shock [10]. The protection of the glycocalyx after hemorrhagic shock is an essential strategy, because an impaired glycocalyx decreases the effect of fluid resuscitation. Thus, novel approaches to minimize glycocalyx damage and improve patient care during hemorrhagic shock are being explored [2,3,4].Although various strategies for the protection of the glycocalyx, such as suppressing inflammatory responses, hypercoagulation, and supplying adequate oxygen [7], have been reported, no fully protective method has been established to date. Therefore, new and effective means of glycocalyx protection are needed.

A previous study indicated that H_2_ inhalation effectively protected the glycocalyx in a rat model of hemorrhagic shock [12]. H_2_ gas inhalation during hemorrhagic shock and resuscitation stabilized post-resuscitation hemodynamics and improved short-term survival in rats. The study [12] found that H_2_ reduced oxidative stress and prevented endothelial glycocalyx damage by inhibiting inflammation-mediated syndecan-1 shedding. H_2_ converts to water molecules by reacting with hydroxyl radicals (OH), which are a highly reactive oxygen species, and inhibits the reaction of these hydroxyl radicals with biological substances. When H_2_ is administered, it diffuses through the membrane and reaches the whole body via the bloodstream. Its effect on the prognosis after hemorrhagic shock has been reported [12,13]. However, even if H_2_ alone can scavenge hydroxyl radicals, it cannot replace the volume equivalent to the blood lost. From this point of view, H_2_-rich saline would be an ideal product for treating massive hemorrhage, as it can decrease the level of hydroxyl radicals and simultaneously replace the blood volume. However, the efficacy of H_2_-rich saline on the glycocalyx remains unexplored. In this study, we revealed that the hemorrhage-induced changes in glycocalyx thickness in the mesentery and quadriceps were significantly improved in the H_2_-NS group. This indicates that H_2_-rich saline has a more protective effect on glycocalyx degeneration after hemorrhagic shock than conventional resuscitation fluids, including normal saline and hydroxyethyl starch. Furthermore, the amount of fluid required for resuscitation was significantly less for H_2_-rich saline than for conventional fluids.

Oxidative stress is induced by hemorrhagic shock, which damages tissues and multiple organs. Free radical damage also critically injures the endothelial glycocalyx [20]. In this study, the amount of ROS was significantly decreased in the H2-NS group, which implies that H_2_ attenuated the increase in ROS after hemorrhagic shock, protecting the glycocalyx.

Previous studies evaluated blood data, such as syndecan-1 or hyaluronic acid levels, and used electron microscope images to assess glycocalyx status [10,21]. These methods provided some evidence; however, these approaches do not reflect the real-time condition of the glycocalyx. GlycoCheck is a non-invasive system that can capture videos of blood flow in capillaries and analyze it using its built-in software [22]. Using the GlycoCheck system, we were able to observe the glycocalyx in vivo and understand real-time changes in glycocalyx thickness.

Additionally, in this study, we showed the beneficial effect of H_2_-rich solutions on ALI after hemorrhagic shock. The effectiveness of H_2_-containing immersion for ischemia–reperfusion injury of the lung has previously been reported. The mechanism of the lung-protective effect of H_2_ was attributed to its anti-inflammatory action [23]. Our results suggested that H_2_ can also decrease oxidative stress, which is another pathway for lung protection. Similar to our results, H_2_-containing solutions have been shown to reduce oxidative stress in a hemorrhagic shock model [24]. ALI is associated with cytokines and antioxidant properties. H_2_ can regulate the related intracellular signals. H_2_-rich saline can prevent increases in damaging cytokines and oxidative stress in a hemorrhagic shock model, thereby protecting against ALI. In addition to this pathway, our results showed that the amount of H_2_-rich saline required for hemorrhagic shock resuscitation was significantly smaller than that required with normal saline. The effects of H_2_ itself and the reduction in the fluid volume used can contribute to the lung protective effects of H_2_-rich saline.

Regarding myocardial contractility, the proper functioning of the endothelial glycocalyx helps maintain vascular elasticity and ensures appropriate blood flow, thereby supporting cardiac function [25]. Conversely, if the glycocalyx is damaged, vascular stiffness ensues, disrupting blood flow and placing additional strain on the heart. Thus, the health of the endothelial glycocalyx is closely related to overall cardiac function, including myocardial contractility. Our results indicated that dP/dt was significantly improved in the H_2_-NS group compared to the shock group. The protective effect of H_2_ on glycocalyx degeneration can contribute to improved cardiac function. A previous study showed that H_2_ gas inhalation attenuated glycocalyx damage and stabilized hemodynamics after hemorrhagic shock [12]. By preserving vascular permeability and cardiac contractility, H_2_ contributes to stable hemodynamic function.

Blood gas analysis also showed the beneficial effects of H_2_-rich solutions. They reduce oxidative stress and lead to improvements in arterial oxygenation and suppress lactate elevation. H_2_-rich solutions may contribute to the less amount of fluid resuscitation and result in better outcomes.

Although we demonstrated the beneficial effects of H_2_-rich saline in fluid resuscitation of hemorrhagic shock in a rat model, our study had some limitations. First, we used only male rats in this study. Sex may influence the effect of H_2_. Therefore, studies including both sexes are needed for further assessments of H_2_-rich solutions. Second, we did not measure the glycocalyx thickness in the lung and heart. The intestinal vasculature is easy to access and observe with the GlycoCheck system; therefore, we focused on the mesentery to evaluate the efficacy of H_2_-rich saline during fluid resuscitation. Third, in the current study, we evaluated the glycocalyx thickness using the GlycoCheck system for real-time analysis, rather than electron microscopy, as has conventionally been used to assess the glycocalyx. While the accuracy of the GlycoCheck system has already reported [26], the number of reports using this system remains limited. Fourth, we only investigated the efficacy of a H_2_-containing solution in the hemorrhagic shock model. H_2_ can also be administered via inhalation, oral ingestion, immersion, and eye drops [27]. Whereas H_2_ gas is flammable in room air, complicating the use of H_2_ inhalation treatment, intravenous administration of H_2_-rich saline is safer and easier to administer at the bedside. Fifth, individual variations affect endothelial glycocalyx thickness. However, due to the study’s design constraints, we compared different groups rather than using the same sample at multiple time points. Sixth, there is a concern regarding the euthanasia of 30 rats at 2 h post-injury and its potential impact on the survival rate results. Although blood transfusion can be effective for patients with critical hemorrhagic shock, its use is associated with some risks. Additionally, sevoflurane for anesthesia is reported to have a protective effect on the glycocalyx via a reduction in oxidative stress [28]. However, all rats in this study used sevoflurane, so we can remove the differences between the groups. In our study, H_2_-rich saline showed improvements in lung injury compared with the other solutions. The study comparing H_2_-rich saline and fresh-frozen plasma is anticipated. Future studies including longer observation periods are needed.

## 5. Conclusions

We have demonstrated the protective effect of H_2_-rich saline on endothelial glycocalyx degeneration in a rat hemorrhagic shock model. H_2_-rich saline improved outcomes in terms of cardiac contractility, the amount of oxidative stress, and ALI. These findings imply that the use of H_2_-rich saline as fluid resuscitation for hemorrhagic shock cases may lead to improved survival rates.

## Figures and Tables

**Figure 1 biomedicines-13-00833-f001:**
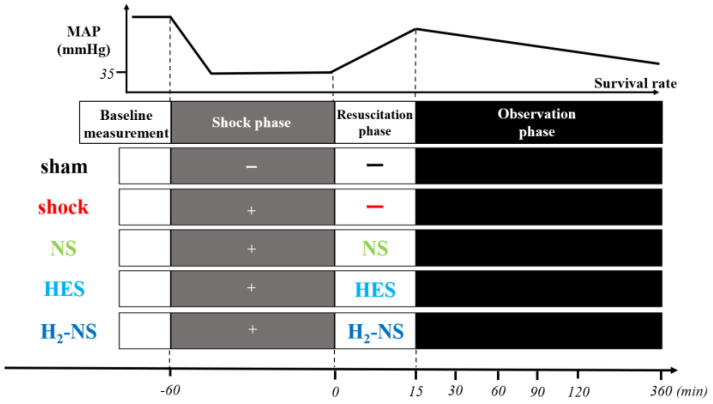
Protocols for experiments. Rats were randomly divided into five groups (*n* = 6 in each group): sham (sham operated), shock (only hemorrhagic shock), NS (hemorrhagic shock + fluid resuscitation using normal saline), HES (hemorrhagic shock + fluid resuscitation using hydroxyethyl starch), and H_2_-NS (hemorrhagic shock + fluid resuscitation using hydrogen-rich saline). Abbreviations: MAP, mean arterial pressure; NS, normal saline; HES, hydroxyethyl starch; H_2_-NS, hydrogen-rich saline.

**Figure 2 biomedicines-13-00833-f002:**
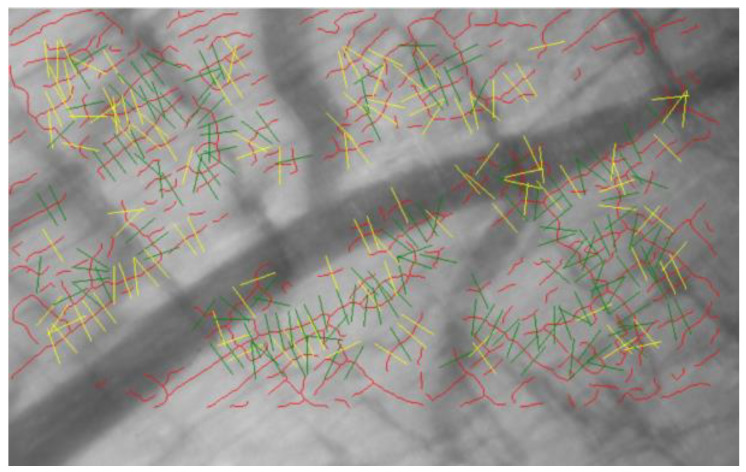
Image captured by the GlycoCheck system. Red lines indicate capillaries. The capillaries are divided into 10 μm sections (green lines). Yellow lines indicate capillaries that do not meet the criteria of the analysis algorithm. After the image is captured, the device software analyzes the microvascular endothelial glycocalyx thickness in real-time.

**Figure 3 biomedicines-13-00833-f003:**
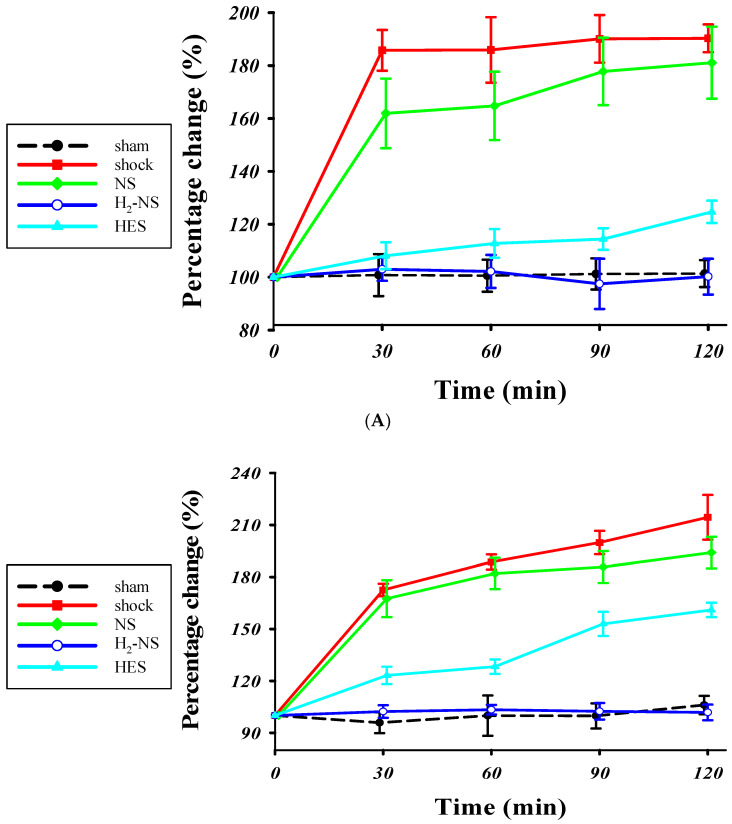
The PBR in the (**A**) mesentery and (**B**) quadriceps femoris. The increase in the PBR means glycocalyx degeneration. The PBR was significantly higher in the shock group than in the sham and H_2_-NS groups at each time point. H_2_-NS prevented the hemorrhagic shock-induced PBR increase in both the mesentery and quadriceps. The H_2_-NS group showed a lower PBR increase than the NS and HES groups, indicating that the glycocalyx is more effectively protected in the mesentery and quadriceps by the hydrogen-containing solution (**A**,**B**). Data are represented as the percentage change from the basal glutamate levels at each point. Values are expressed as the median and interquartile range. *p* values were adjusted with Tukey’s post hoc test. * *p* < 0.05. Abbreviations: PBR, perfusion boundary region; NS, normal saline; HES, hydroxyethyl starch; H_2_-NS, hydrogen-rich saline.

**Figure 4 biomedicines-13-00833-f004:**
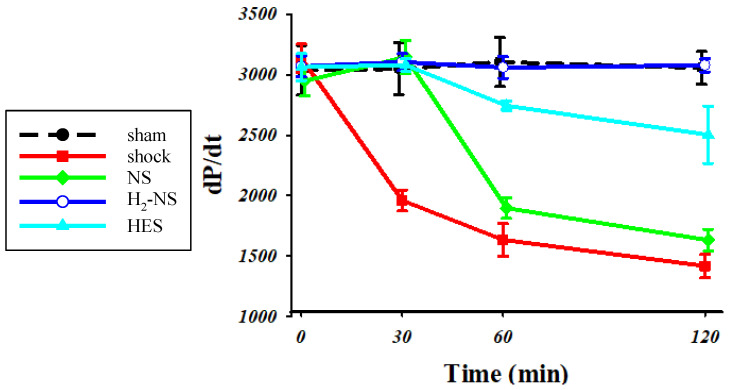
For assessments of cardiac contraction, we measured dP/dt. The animals in the shock group showed significantly lower dP/dt than those in the sham group. The H_2_-NS group attenuated the decrease in dP/dt compared to the shock group. Although the dP/dt in the NS and HES groups was decreased over the course of time, the dP/pt in the H_2_-NS group was significantly maintained after hemorrhagic shock. Values are expressed as the median and interquartile range. *p* values were adjusted with Tukey’s post hoc test. * *p* < 0.05. Abbreviations: NS, normal saline; HES, hydroxyethyl starch; H_2_-NS, hydrogen-rich saline.

**Figure 5 biomedicines-13-00833-f005:**
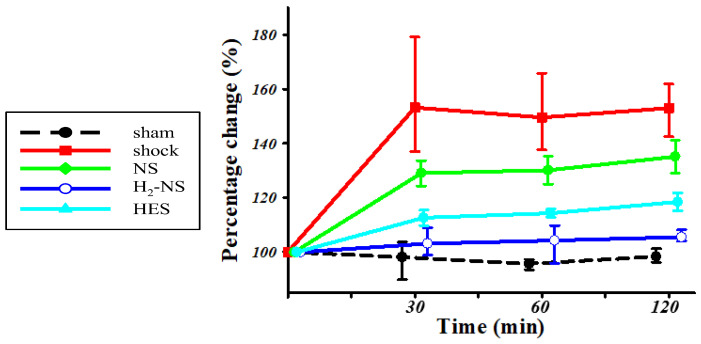
The amount of hydroperoxides in the blood serum. The amount of hydroperoxides in the blood serum obtained from the shock group was significantly greater than that obtained from the sham group at all time points. The amount of hydroperoxides in the blood serum obtained from the H_2_-NS group indicated no significant difference compared to that of the sham group at all time points. *p* values were adjusted with Tukey’s post hoc test. * *p* < 0.05. Abbreviations: NS, normal saline; HES, hydroxyethyl starch; H_2_-NS, hydrogen-rich saline.

**Figure 6 biomedicines-13-00833-f006:**
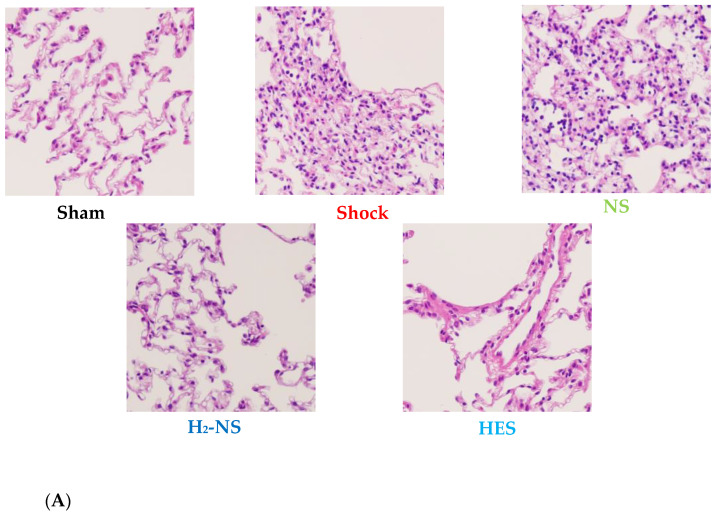
Histopathological analysis of a section from the left upper lung lobe. The section was stained with hematoxylin and eosin and was observed at 20× magnification under a light microscope (**A**). Acute lung injury was evaluated using the ALI score. The ALI score was significantly higher in the shock group than in the sham group and was significantly lower in the H_2_-NS group than in the shock group (**B**). *p* values were adjusted with Tukey’s post hoc test. * *p* < 0.05. Abbreviations: NS, normal saline; HES, hydroxyethyl starch; H_2_-NS, hydrogen-rich saline.

**Figure 7 biomedicines-13-00833-f007:**
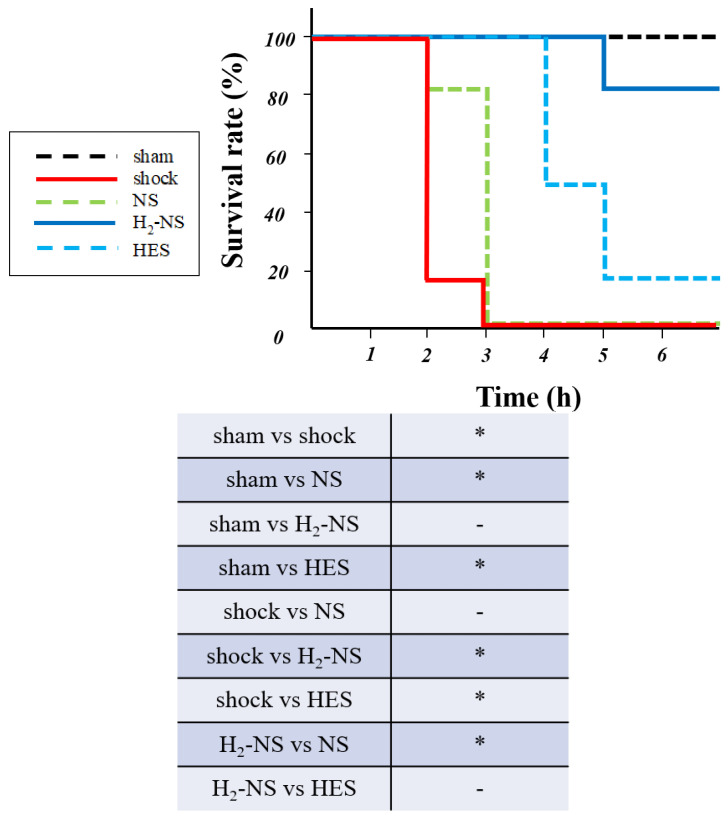
Survival rate at 6 h after fluid resuscitation. The shock and NS groups demonstrated a significantly lower survival rate than the other groups. Hydrogen contributed to the improvement in the survival rate after hemorrhagic shock. *p* values were adjusted with Tukey’s post hoc test. * *p* < 0.05. Abbreviations: NS, normal saline; HES, hydroxyethyl starch; H_2_-NS, hydrogen-rich saline.

**Table 1 biomedicines-13-00833-t001:** Characteristics of the rats.

	Sham	Shock	NS	HES	H_2_-NS	*p* Values
**Body weight (g)**	501.7 ± 14.7	498.3 ± 14.7	495.0 ± 10.5	496.7 ± 16.3	498.3 ± 18.3	0.941
**Blood loss (mL)**	-	10.0 ± 0.3	10.0 ± 0.3	10.0 ± 0.2	10.0 ± 0.3	0.938
**Resuscitation fluid volume (mL)**	-	-	31.7 ± 1.4	10.3 ± 0.7	9.9 ± 0.6	<0.001 (H_2_-NS vs. NS) 0.781 (H_2_-NS vs. HES)

Abbreviations: NS, normal saline; HES, hydroxyethyl starch; H_2_-NS, hydrogen-rich saline.

**Table 2 biomedicines-13-00833-t002:** Arterial blood gas at different time points.

	Sham	Shock	NS	HES	H_2_-NS
**Baseline**					
**pH**	7.40 ± 0.03	7.18 ± 0.03	7.28 ± 0.03	7.35 ± 0.03	7.37 ± 0.03
**PaCO_2_ (mmHg)**	33.7 ± 1.03	33.7 ± 1.75	34.0 ± 2.1	33.7 ± 1.5	33.0 ± 1.1
**PaO_2_ (mmHg)**	232.7 ± 11.38	233.0 ± 8.32	240.17 ± 8.33	234.5 ± 9.31	234.2 ± 7.89
**BE (mEq/L)**	−1.1 ± 0.24	−1.05 ± 0.33	−1.17 ± 0.31	−0.98 ± 0.25	−1.0 ± 0.36
**Lac (mmol/L)**	1.75 ± 011	1.75 ± 0.13	1.7 ± 0.11	1.65 ± 0.12	1.8 ± 0.11
**Hb (g/dL)**	13.61 ± 0.48	13.63 ± 0.43	13.65 ± 0.54	13.67 ± 0.61	13.57 ± 0.55
**BUN (mg/dL)**	27.0 ± 1.27	27.5 ± 1.38	27.67 ± 1.21	27.17 ± 1.72	26.5 ± 1.05
**Cre (mg/dL)**	0.47 ± 0.1	0.43 ± 0.1	0.43 ± 0.1	0.48 ± 0.08	0.47 ± 0.1
**Sodium (mEq/L)**	138.3 ± 1.97	138.5 ± 2.07	138.5 ± 3.08	139.5 ± 3.27	138.85 ± 3.55
**Potassium (mEq/L)**	3.92 ± 0.29	3.77 ± 0.24	3.87 ± 0.25	3.88 ± 0.26	3.83 ± 0.25
**30 min after resuscitation**					
**pH**	7.402 ± 0.04	7.18 ± 0.03 *^†^	7.28 ± 0.02 *^†^	7.35 ± 0.03 *	7.37 ± 0.03
**PaCO_2_ (mmHg)**	33.7 ± 1.76	32.7 ± 1.37	32.0 ± 1.27	31.8 ± 0.75	32.8 ± 0.6
**PaO_2_ (mmHg)**	234.2 ± 5.56	96.3 ± 4.07 ^†^	87.6 ± 1.22 *^†^	166.9 ± 6.50	230.9 ± 12.36
**BE (mEq/L)**	−1.15 ± 0.39	−13.63 ± 0.83 *^†^	−6.4 ± 0.4 *	−4.8 ± 0.37	−3.77 ± 0.85
**Lac (mmol/L)**	1.73 ± 0.11	10.93 ± 0.69 *^†^	8.08 ± 0.23 *	4.05 ± 0.34	3.05 ± 0.17
**Hb (g/dL)**	13.65 ± 0.41 ^†^	6.97 ± 0.22 *^†^	6.77 ± 0.22 *^†^	9.2 ± 0.54 *^†^	9.92 ± 0.12*
**BUN (mg/dL)**	27.17 ± 1.72	32.67 ± 1.37 *^†^	28.33 ± 1.5	27.83 ± 1.84	26.7 ± 0.82
**Cre (mg/dL)**	0.47 ± 0.12	2.27 ± 0.18 *^†^	1.33 ± 0.14 *^†^	1.38 ± 0.12 *^†^	0.53 ± 0.1
**Sodium (mEq/L)**	139.7 ± 3.6	139.3 ± 2.8	140.0 ± 3.0	140.0 ± 3.3	138.5 ± 2.2
**Potassium (mEq/L)**	3.95 ± 0.24	4.4 ± 0.14 *^†^	4.0 ± 0.3	3.87 ± 0.22	3.93 ± 0.21
**120 min after resuscitation**					
**pH**	7.43 ± 0.03	7.22 ± 0.03 *^†^	7.24 ± 0.02 *^†^	7.36 ± 0.03 *	7.39 ± 0.02
**PaCO_2_ (mmHg)**	32.0 ± 1.41	31.67 ± 1.86	32.5 ± 2.07	32.33 ± 1.75	32.5 ± 1.87
**PaO_2_ (mmHg)**	232.67 ± 11.38	95.48 ± 3.20 *^†^	75.10 ± 2.80 *^†^	168.10 ± 3.80 *^†^	232.82 ± 8.22
**BE (mEq/L)**	−1.70 ± 0.37 ^†^	−16.13 ± 0.48 *^†^	−11.63 ± 0.86 *^†^	−6.40 ± 0.44 *^†^	−4.10 ± 0.76 *
**Lac (mmol/L)**	1.72 ± 0.10 ^†^	12.67 ± 0.90 *^†^	8.70 ± 0.48 *^†^	5.77 ± 0.46 *^†^	3.40 ± 0.33 *
**Hb (g/dL)**	13.48 ± 0.46	6.83 ± 0.22 *^†^	7.02 ± 0.50 *^†^	9.82 ± 0.25 *^†^	10.6 ± 0.41 *
**BUN (mg/dL)**	26.83 ± 1.47	32.0 ± 1.41 *^†^	29.33 ± 2.07 *	28.67 ± 0.52	27.83 ± 1.17
**Cre (mg/dL)**	0.52 ± 0.09	5.78 ± 0.28 *^†^	2.4 ± 0.15 *	2.45 ± 0.22 *	0.85 ± 0.1
**Sodium (mEq/L)**	140.17 ± 3.19	141.5 ± 4.76	137.0 ± 1.41	140.33 ± 3.01	138.33 ± 3.27
**Potassium (mEq/L)**	3.88 ± 0.17	4.17 ± 0.18 *	3.93 ± 0.1	3.88 ± 0.15	3.82 ± 0.17

* *p* value vs. sham < 0.05. ^†^ *p* value vs. H_2_-NS < 0.05 Abbreviations: NS, normal saline; HES, hydroxyethyl starch; H_2_-NS, hydrogen-rich saline; BE, base excess; Lac, lactate; Hb, hemoglobin; BUN, blood urea nitrogen; Cre, creatinine.

## Data Availability

The original contributions presented in this study are included in the article. Further inquiries can be directed to the corresponding author.

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
