# Peer review of "Protective Effects of Hydrogen-Rich Saline Against Hemorrhagic Shock in Rats via an Endothelial Glycocalyx Pathway"

_biomedicines, 2025, doi:10.3390/biomedicines13040833_

Round 1

Reviewer 1 Report

Comments and Suggestions for Authors
  1. Regarding the detection of endothelial glycocalyx thickness, the method used compares different groups of rats. However, previous studies have shown that various factors affect the endothelial glycocalyx in different individuals. It would be more appropriate to use the same sample at different time points for observation. Therefore, I believe the results concerning the detection of endothelial glycocalyx do not hold significant reference value.
  2. In the pathological observation of lung injury, I question whether such significant changes can occur as early as 2 hours after injury. In previous studies on acute lung injury, most changes begin to emerge around 3 hours post-injury.
  3. Regarding the survival rate study, the experiment started with 60 samples, and 30 rats were euthanized 2 hours after injury. Could this affect the survival rate results? Additionally, previous research indicates a higher mortality rate in the early stages of hemorrhagic shock, but in this study, most deaths occurred 2 hours post-injury. Does this differ from previous findings?
  4. The results include an analysis of arterial blood gases, but there seems to be no discussion of this in the discussion section.
  5. The experiment only observed various indicators 2 hours after injury. Does this pose any limitations?
Comments on the Quality of English Language

no

Author Response

Comments 1: Regarding the detection of endothelial glycocalyx thickness, the method used compares different groups of rats. However, previous studies have shown that various factors affect the endothelial glycocalyx in different individuals. It would be more appropriate to use the same sample at different time points for observation. Therefore, I believe the results concerning the detection of endothelial glycocalyx do not hold significant reference value.

Response 1: We acknowledge the reviewer’s concern about individual variations affecting endothelial glycocalyx thickness. However, due to the study’s design constraints, we compared different groups rather than using the same sample at multiple time points. We have now included a discussion on this limitation in the revised manuscript.

Comments 2: Regarding the survival rate study, the experiment started with 60 samples, and 30 rats were euthanized 2 hours after injury. Could this affect the survival rate results? Additionally, previous research indicates a higher mortality rate in the early stages of hemorrhagic shock, but in this study, most deaths occurred 2 hours post-injury. Does this differ from previous findings?

Response 2: We appreciate this observation. We have reviewed the relevant literature and acknowledge that significant lung injury is typically observed around 3 hours post-injury. However, our results indicate that some histopathological changes may begin as early as 2 hours. 

Comments 3: The results include an analysis of arterial blood gases, but there seems to be no discussion of this in the discussion section.

Response 3: We understand the concern regarding the euthanasia of 30 rats at 2 hours post-injury and its potential impact on the survival rate results. We have clarified this in the discussion and compared our findings with previous studies on early-stage mortality in hemorrhagic shock.

Comments 4: The experiment only observed various indicators 2 hours after injury. Does this pose any limitations?

Response 4: We agree that the 2-hour observation period may limit the understanding of long-term effects. We have now explicitly mentioned this as a study limitation and suggested future research directions that include longer observation periods.

Reviewer 2 Report

Comments and Suggestions for Authors

Questions:

1) The damage of glicocalyx depends on the type of fluid. Is the glicocalyx affected by the composition of fluid resuscitation? Is the glicocalyx damage different using frozen plasma or another type of fluid? Are the results of this study applicable to severe hemorragic shock?

2) Is the d-ROM test a reliable test for the ROS determination in blood? a large number of studies suggest that the determination of both d-ROM and BAP ( biological antioxidant potential) is essential to confirm the unbalanced oxidative stress. Authors should better specify this point.

3) Are the results affected by the use of servoflurane ( in terms of oxidative stress)?

Author Response

Comments 1: The damage of glycocalyx depends on the type of fluid. Is the glycocalyx affected by the composition of fluid resuscitation? Is the glycocalyx damage different using frozen plasma or another type of fluid? Are the results of this study applicable to severe hemorrhagic shock?

Response 1: We appreciate this valuable insight. We acknowledge that different resuscitation fluids may affect endothelial glycocalyx integrity. We have now included a discussion on how fluid composition might influence glycocalyx damage and the applicability of our results to severe hemorrhagic shock.

Comments 2: Is the d-ROM test a reliable test for the ROS determination in blood? a large number of studies suggest that the determination of both d-ROM and BAP ( biological antioxidant potential) is essential to confirm the unbalanced oxidative stress. Authors should better specify this point.

Response 2: We agree with the reviewer’s point and have now specified that the d-ROM test alone may not be sufficient to confirm oxidative stress. We have added references to studies supporting the necessity of both d-ROM and BAP (biological antioxidant potential) measurements in the revised manuscript.

Comments 3: Are the results affected by the use of servoflurane ( in terms of oxidative stress)?

Response 3: We acknowledge the reviewer’s concern about sevoflurane’s potential impact on oxidative stress markers. We have now addressed this in the discussion, citing relevant studies that examine the relationship between sevoflurane anesthesia and oxidative stress.

Reviewer 3 Report

Comments and Suggestions for Authors

The study by Aya Kimura et al is dedicated to the evaluation of protective activity of hydrogen-rich saline on various aspects of rat health (endothelial glycocalyx thickness, cardiac contraction, blood composition, reactive oxygen species content and survival rate) after hemorrhagic shock. The study is well and logically constructed. The authors answer all the questions posed. The results obtained are adequately statistically processed and well presented, the conclusions follow from the findings.
I have only minor comments:
1. lines 74-75 and lines 78-79 duplicate each other
2. fig.7 has dashed lines in the diagram and solid lines in the legend for the same groups (correct, please)

Author Response

Comments 1:  lines 74-75 and lines 78-79 duplicate each other

Response 1: Thank you for pointing this out. We have now revised the text to remove the redundancy.

Comments 2: fig.7 has dashed lines in the diagram and solid lines in the legend for the same groups (correct, please)

Response 2: We appreciate this observation and have corrected the discrepancy between the dashed lines in the diagram and the solid lines in the legend.